# Transcriptomics and RNA-Based Therapeutics as Potential Approaches to Manage SARS-CoV-2 Infection

**DOI:** 10.3390/ijms231911058

**Published:** 2022-09-21

**Authors:** Cristian Arriaga-Canon, Laura Contreras-Espinosa, Rosa Rebollar-Vega, Rogelio Montiel-Manríquez, Alberto Cedro-Tanda, José Antonio García-Gordillo, Rosa María Álvarez-Gómez, Francisco Jiménez-Trejo, Clementina Castro-Hernández, Luis A. Herrera

**Affiliations:** 1Unidad de Investigación Biomédica en Cáncer, Instituto Nacional de Cancerología-Instituto de Investigaciones Biomédicas, Universidad Nacional Autónoma de México (UNAM), Avenida San Fernando No. 22 ColC. Sección XVI, Tlalpan. C.P., Mexico City 14080, Mexico; 2Genomics Laboratory, Red de Apoyo a la Investigación, Universidad Nacional Autónoma de México, Vasco de Quiroga 15, Belisario Domínguez Secc 16, Tlalpan, Mexico City 14080, Mexico; 3Instituto Nacional de Medicina Genómica, Periférico Sur 4809, Arenal Tepepan, Tlalpan. C.P., Mexico City 14610, Mexico; 4Oncología Médica, Instituto Nacional de Cancerología, Avenida San Fernando No. 22 Col. Sección XVI, Tlalpan. C.P., Mexico City 14080, Mexico; 5Clínica de Cáncer Hereditario, Instituto Nacional de Cancerología, Avenida San Fernando No. 22 Col. Sección XVI, Tlalpan. C.P., Mexico City 14080, Mexico; 6Instituto Nacional de Pediatría, Insurgentes Sur No. 3700-C, Coyoacán. C.P., Mexico City 04530, Mexico

**Keywords:** SARS-CoV-2, transcriptomics, RNA therapeutics, precision medicine, precision public health

## Abstract

SARS-CoV-2 is a coronavirus family member that appeared in China in December 2019 and caused the disease called COVID-19, which was declared a pandemic in 2020 by the World Health Organization. In recent months, great efforts have been made in the field of basic and clinical research to understand the biology and infection processes of SARS-CoV-2. In particular, transcriptome analysis has contributed to generating new knowledge of the viral sequences and intracellular signaling pathways that regulate the infection and pathogenesis of SARS-CoV-2, generating new information about its biology. Furthermore, transcriptomics approaches including spatial transcriptomics, single-cell transcriptomics and direct RNA sequencing have been used for clinical applications in monitoring, detection, diagnosis, and treatment to generate new clinical predictive models for SARS-CoV-2. Consequently, RNA-based therapeutics and their relationship with SARS-CoV-2 have emerged as promising strategies to battle the SARS-CoV-2 pandemic with the assistance of novel approaches such as CRISPR-CAS, ASOs, and siRNA systems. Lastly, we discuss the importance of precision public health in the management of patients infected with SARS-CoV-2 and establish that the fusion of transcriptomics, RNA-based therapeutics, and precision public health will allow a linkage for developing health systems that facilitate the acquisition of relevant clinical strategies for rapid decision making to assist in the management and treatment of the SARS-CoV-2-infected population to combat this global public health problem.

## 1. Introduction

The genome of SARS-CoV-2 consists of a single-stranded positive-sense RNA with a sequenced genome of approximately 29.9 Kb in size [1]. In a comprehensive study conducted by Kim and collaborators, a high-resolution map of the SARS-CoV-2 transcriptome was analyzed using the ‘‘Sequencing-By-Synthesis (SBS)’’ and direct RNA sequencing (DRS) approaches, which showed that it produces RNAs encoding unknown ORFs and at least 41 potential RNA modification sites [2]. Currently, transcriptomics has become an important tool in the study of SARS-CoV-2 in order to generate new insights into the viral genome to identify new therapeutic targets. An example of this is the Global Initiative on Sharing All Influenza Data (GISAID) platform, which has submitted 12.2 million SARS-CoV-2 genomes and where public information has been supported to develop tools for genomic analysis [3], development of vaccines [4], and diagnostic tests for epidemiological surveillance of SARS-CoV-2 [5], which serves to monitor the emergence of new viral variants and helps to comprehend how viruses evolve and spread in pandemics.

On the other hand, RNA therapeutics have gained much attention with the development of new vaccines based on RNA molecules. This field began in 1978 and has taken a radical turn, and now there is a bridge between the use of precision medicine and RNA-based therapies [6], where transcriptomics and knowledge of the SARS-CoV-2 viral genome play important roles in the development of new targeted drugs and therapies that are important tools for developing precision medicine (PM) in this field.

Herein, we provide an overview of the impact of transcriptomics studies and how this technology has influenced the generation of new knowledge regarding SARS-CoV-2. In addition, we provide the state of the art of RNA-based therapeutics with a focus on the relationship to SARS-CoV-2. Furthermore, we describe possible strategies for the development of new vaccines and treatments that can be applied in infected patients. Moreover, the importance of precision public health (PPH) in the management of patients infected with SARS-CoV-2 and its relationship to the information generated by transcriptomics is discussed. Hence, using transcriptomics, RNA-based therapeutics, and population health registry data serve to establish a merger between technologies to create synergism that allows for the development of advanced systems focused on PPH that facilitate identification of relevant clinical strategies for rapid decision making and support tailored management of patients infected with SARS-CoV-2.

## 2. The Use of Transcriptomics as a Tool for Understanding SARS-CoV-2 Biology

As a concept, the transcriptome is defined as the complete set of transcripts in a cell and their quantity for a specific developmental stage, particular cell, or tissue type [7]. This term is also used to describe all transcripts synthesized under particular physiological conditions such as SARS-CoV-2 infection, in which the viral transcriptome is mixed with the host organism’s transcriptome, since its transcriptional machinery allows replication of SARS-CoV-2 viral particles [8,9]. In particular, SARS-CoV-2 studies relating to transcriptome analysis have two primary research areas: study of the viral sequences of SARS-CoV-2 and analysis of the intracellular signaling pathways that regulate infection and pathogenesis of COVID-19 [10].

Analysis of the SARS-CoV-2 transcriptome has allowed the identification of new sequences and their genetic variations. However, the functions of these sequences are currently unknown, so it has not been possible to establish their importance in the process of viral infection with SARS-CoV-2 [11]. Moreover, since the beginning of the pandemic, new variants of SARS-CoV-2 such as the Alpha (B.1.1.7) [12], Beta (B.1.351) [13], Delta (B.1.617.2) [14], and Omicron variants [15] have been reported, among others [16,17,18,19,20,21], which have been considered of clinical importance because they are related to pathological characteristics such as transmissibility immune evasion and severity of the respiratory disease [22]. Therefore, their detection and analysis have been relevant to clinical and sociopolitical decision making to control disease transmission and the associated mortality (Figure 1A). One workflow that has allowed the identification of these new variants is COVIDSeq, which was developed by Bhoyar and collaborators. It is based on the Illumina platform consisting of multiplex PCR and barcode-based sequencing analysis, which allowed the identification of 1143 unique variants of SARS-CoV-2 with high sensitivity and specificity (~97%) [23]. These results showed that the analysis of SARS-CoV-2 variants using transcriptomics is a useful diagnostic tool in the epidemiological monitoring of SARS-CoV-2.

### 2.1. Viral Variants and Quasispecies for SARS-CoV-2

SARS-CoV-2 genome characterization enabled the identification of viral particles within the host that can develop stable variations of the viral genome defined by a spectrum of mutations, which were defined as quasispecies, whose development depends on the processes of viral replication and host interaction [24]. The selection of quasispecies acts on the set of viral particles, which confers viral adaptability and stability, allowing greater transmissibility between hosts, as well as improving the mechanisms of immune response evasion [25]. Chaudry and collaborators described SARS-CoV-2 quasispecies production in different cell lines by serial passages, showing that in Vero E6 cells, mutation of the FURIN cleavage site in protein S, which is related to the priming or activation of SARS-CoV-2 spike proteins, makes SARS-CoV-2 competent for membrane fusion and infection [26,27]. However, serial passages in the Calu-3 and Caco-2 cell lines showed that some quasispecies only obtained variations in protein E. These results indicated that SARS-CoV-2 infection can differentially generate stable quasispecies among infected cell lines, conveying adaptability for viral replication and active infection [26], which was demonstrated by Al Khatib and collaborators, who sequenced the transcriptome of nasopharyngeal samples from patients with moderate disease and severe disease infected with SARS-CoV-2 to evaluate whether changes in the viral genotype were associated with COVID-19 severity. This study allowed the identification of 236 SARS-CoV-2 quasispecies, which suggested that variability in the viral genotype is associated with severe respiratory disease. Particularly, the intrahost analysis of these viral variants demonstrated the association of the mutations in protein S and ORF1ab with severe disease development in patients [28], corroborating that the development of intrahost quasispecies is a mechanism of adaptability of SARS-CoV-2.

### 2.2. Metabolic Pathways Defined by Transcriptome Analysis in SARS-CoV-2 Infection

Transcriptomic analysis of SARS-CoV-2-infected individuals has identified intracellular signaling pathways associated with the infection process [29] and host pathogenesis [30] through differential gene expression analysis approaches (Figure 1B) involving the identification of genes actively transcribed in response to viral infection compared to an uninfected individual [31], which has made possible the development and repurposing of drugs for use in therapies with increased likelihood of success for the treatment of COVID-19 [32]. For example, the drug repositioning analysis by Krishnamoorthy and collaborators identified intracellular signaling pathways such as neuroactive ligand–receptor interactions that are deregulated in infection with SARS-CoV-2 virus in the Calu-3 epithelial cell line, while a Cogena drug repositioning analysis identified target proteins in this signaling pathway for valproic acid, positioning it as an alternative treatment for COVID-19 [33]. Likewise, Salgado-Albarrán and collaborators suggested that epidrugs are alternatives for the treatment of COVID-19 after identifying differentially expressed epigenes in respiratory disease caused by SARS-CoV-2 such as BRD4, which is a chromatin reader protein involved in the immune response [32,34] that is a target of the epidrug fedratinib [32]. In addition, an analysis of the transcriptome of the A549 and HepG2 epithelial cell lines treated with exosomes extracted from the plasma of patients with COVID-19 revealed that SARS-CoV-2 infection causes upregulation of the NOTCH signaling pathway related to hyperinflammation in COVID-19 patients [35]. In this case, analysis of the transcriptomes of patients with COVID-19 has allowed for the identification of new intracellular signaling pathways involved in the development of severe respiratory disease in the host, which has applications in the prognosis of patients with an active viral infection.

However, there are still gaps in identifying new therapeutic targets, since it is not clear which genes could be actionable targets for drugs in COVID-19 treatment due to the clinical sample being considered homogenous. A novel approach that can address these variables is spatial transcriptomics, which analyzes the transcriptome related to specific locations in tissues as single cells, capture points, or delimited areas within a tissue [36], as was demonstrated in COVID-19 spatial transcriptomic studies by Margaroli and collaborators of lung tissue autopsies of patients with alveolar damage caused by SARS-CoV-2. In this study, they described a gene expression profile that included genes related to vasculopathy such as MSRB2, genes related to alveolar epithelial phenotype and type II pneumocytic metaplasia such as FGFR3, genes related to epithelial-mesenchymal transition such as CDH1, and genes associated with differential macrophage activation such as MS4A4A [37], which have already been identified as novel findings in the research of COVID-19 pathogenesis [38]. Therefore, new experimental approaches such as spatial transcriptomics have made it possible to broaden the understanding of the process of infection by SARS-CoV-2, in addition to the mechanisms associated with its pathogenicity.

### 2.3. New RNA-Seq Pipelines for SARS-CoV-2 Research

New RNA-sequencing (RNA-seq) bioinformatics strategies have been developed to improve the processing of transcriptome sequencing results in order to identify new associations between the host and SARS-CoV-2, such as the Dual RNA-Seq Analysis Pipeline (dRAP), which allows for the simultaneous identification of differentially expressed genes in the host and pathogen genes while optimizing processing of the sample of patients infected with COVID-19 [10], indicating the cost–benefit of this methodology (Figure 1C). Additionally, a tool that was developed by Liu and collaborators, CoV-Seq, allows for the analysis of new SARS-CoV-2 variants in user-friendly interfaces, making it accessible to users not specialized in programming [3].

Significantly, the development of public databases and platforms dedicated to the collection and analysis of sequencing results of the transcriptome associated with SARS-CoV-2 allows access to the raw data by researchers around the world who are not required to be experts in programming [39]. For example, the RECoVERY [40] platform allows the user to analyze public sequencing data or their own massive sequencing data, and the SARSCOVIDB [41] and COVID-19 CG [42] platforms provide the user with molecular information from public RNA-seq data about genes that are differentially expressed in patients with SARS-CoV-2 infection, as well as the analysis of variations and their association with clinical variables. Above all, these new research platforms provide updated resources for the analysis and identification of novel information regarding SARS-CoV-2 research.

## 3. Single-Cell Transcriptomics in SARS-CoV-2 Research

In recent months, single-cell RNA sequencing (scRNAseq) has been harnessed to further our biological understanding of COVID-19 infection in response to the current global pandemic. The scRNAseq method reveals complex and rare cell populations, uncovers regulatory relationships between genes, and tracks the trajectories of distinct cell lineages in development [43]. These procedures began with single-cell qPCR, performed for the first time on a small number of genes. This was followed by a jump to ~1000 cells that could be analyzed in large-scale studies using integrated fluidic circuits [44,45]. The most common approach is to use microscopic droplets to isolate many cells and then sequence the libraries relatively shallowly. To identify which cell a given transcript came from, the use of cellular barcodes has increased throughput even further to hundreds of thousands of cells, with the latest developments in spatial methods integrating the spatial location of transcriptomic information within tissue sections [46]. This high-throughput, low-depth paradigm is typical for experiments using the popular 10X Genomics Chromium platform [47,48]. This platform is currently able to detect 500–1500 genes per primary cell [49]. Thus, scRNAseq has provided information on the coordinated response to SARS-CoV2 viral infections at the single-cell level, making this technology an important tool for the identification of new biomarkers for the diagnosis and prognosis of SARS-CoV-2 infections.

### 3.1. Pipelines for scRNAseq Data Analysis

The rapid development of novel single-cell technologies, most notably multiomics methods that can profile more than one aspect of a cell and methods that provide spatial information, will require novel computational methods to take full advantage of the data [46]. An scRNAseq experiment generates a larger volume of high-dimensional raw data and identifies fewer transcripts than bulk RNA-seq [50]. The steps of a typical scRNAseq analysis include preprocessing (quality control, normalization, data correction, feature selection, and dimensionality reduction) and cell- and gene-level downstream analysis [51]. Some of the software used includes Cell Ranger, which is a set of analysis pipelines that process Chromium single-cell data to align reads, generate feature barcode matrices, and perform clustering and other secondary analyses [52]. Seurat is an R package designed for QC, analysis, and exploration of scRNAseq data [53] (Table 1). Seurat aims to enable users to identify and interpret sources of heterogeneity from single-cell transcriptomic measurements and to integrate diverse types of single-cell data [54]. However, scRNAseq generates multiple errors when various cells are labeled with the same barcode. Scrublet is a method for predicting the effects of multiples (two or more cells are captured within the same reaction, generating a hybrid transcriptome) in analyses and identifying problematic multiplets [55]. Droplet-based scRNAseq methods produce counts of unique molecular identifiers (UMIs) for genes in thousands of cells [56]. The observed counts arise from a mixture of mRNAs produced by the captured cell and those present due to background contamination [45]. SoupX aims to remove cell-free mRNA molecules from each cell to recover the true molecular abundance of each gene in each cell [57]. These methods are crucial to defining cell states and types from data, revealing marker genes and identifying developmental trajectories that relate cells to each other [44], and due to our incomplete knowledge of the biology and pathogenesis of SARS-CoV-2, such bioinformatics tools have great potential for clinical application.

### 3.2. Application of scRNAseq in SARS-CoV-2 Research

The scRNAseq method is powerful at dissecting immune responses and has been applied in COVID-19 studies [67]. To date, most studies on COVID-19 have focused on understanding SARS-CoV-2 infection at the tissue and organ levels [68]. However, viral infection and the response to it are events that initiate at the single-cell level, and various cell types (e.g., lung epithelial or gastrointestinal), even those that do not express the molecular machinery needed for effective infection, may play key roles in viral pathology [67]. The variable host immune response to infection can result in a range of clinical outcomes spanning from an asymptomatic state to severe illness, organ failure, and death [69]. Unlocking the identity of the specific cells involved in infection can clarify transmission patterns, pathogenesis, and risk differences between individuals and will be key to developing effective therapeutic approaches [68].

Recently, a consortium was formed to characterize the immune properties of COVID-19 that consisted of researchers from 39 institutes from different regions of China [67] and the Single Cell Consortium for COVID-19. They generated an scRNAseq dataset and created a comprehensive immune landscape of 1.46 million cells. This project enabled us to identify that different peripheral immune subtype changes are associated with distinct clinical features, including age, sex, severity, and disease stages of COVID-19 [67]. In addition, a portal has been created (https://toppcell.cchmc.org, accessed on 13 September 2022) for single-cell data from COVID-19 patients from eight public datasets, which enables the querying of candidate molecules and pathways in each of these processes [70]. Several studies have been performed using scRNAseq in which it was determined that in patients with COVID-19, epithelial cells exhibit increased expression of the SARS-CoV-2 entry receptor ACE2, which correlated with interferon signals by immune cells. However, critical cases exhibited stronger interactions between epithelial and immune cells, as indicated by ligand receptor expression profiles and activated immune cells, including inflammatory macrophages expressing CCL2, CCL3, CCL20, CXCL1, CXCL3, CXCL10, IL8, IL1B, and TNF [71]. Moreover, the analysis of ACE2 with viral entry-associated protease TMPRSS2 and infection-associated genes (e.g., ANDEP, DPP4) in healthy human individuals across a range of tissues provided insights into potential sites of viral transmission [68,72]. In addition, an scRNAseq study revealed cell-type-specific associations of age, sex, and smoking with overexpression of ACE2, TMPRSS2, and CTSL. The expression of entry factors increased with age and in males, including in airway secretory cells and alveolar type 2 cells [73]. Likewise, they explored the dynamics of the adaptive immune response in asymptomatic close contacts and COVID-19-infected patients and reported that direct asymptomatic contacts exhibited decreased CD4+ naive T cells with a concomitant increase in CD4+ memory and CD8+ TEMRA cells along with expanded clonotypes compared to infected patients. Noticeable proportions of class-switching memory B cells were also observed. Overall, these findings provided insight into the nature of protection in asymptomatic contacts [74]. This type of study provides a valuable resource that is exploitable for translational studies and is a template for future integrative analysis of single-cell datasets from individuals with COVID-19 worldwide (Figure 2).

As spatial transcriptomic technologies reach single-cell resolution, they hold a great deal of promise for the study of cells in tissues with a complex morphology [44,75]. The continuous innovation of scRNAseq technologies and concomitant advances in bioinformatics approaches will greatly facilitate biological and clinical research [73]. In fact, scRNAseq provides new insight into intra- and intercellular communications and network interactions and is a new emerging area that can facilitate the discovery of disease-specific biomarkers and target-based precision therapies [76] (Table 2). Lastly, scRNAseq is revolutionizing our fundamental understanding of biology, and this technique has opened new frontiers of research that go beyond descriptive studies of cellular states. These studies will help us to understand the role of immune cells in COVID-19 pathology and their response to potential treatment regimens.

## 4. Next-Generation Sequencing Platforms for SARS-CoV-2 Genome Research

RNA-seq can be leveraged to study many aspects of RNA biology such as expression, splicing, gene fusion detection, and structure [77]. Currently, innovations in RNA-seq, such as direct RNA-seq [78], detection of long reads [79], spatialomics [80], and the development of new computational tools (TALON for transcriptome quantification [81] or FLAIR for transcriptome assembly [82]) for data analysis have contributed to a complete understanding of the biology of the transcriptome in biomedical research.

Managing the COVID-19 pandemic has required genomic surveillance using second- and third-generation sequencing (STGS), which provides a practical and accurate way to identify new SARS-CoV-2 variants and prevent their spread. The STGS promoted the characterization of variants of interest (VOIs) and variants of concern (VOCs) to prioritize monitoring and viral research and to inform the ongoing response to the COVID-19 pandemic [83,84].

Globally, systems have been established to detect potential VOIs or VOCs and SARS-CoV-2 genomes have been deposited in worldwide platforms of genomic surveillance; for example, GISAID has encouraged and facilitated the sharing of COVID-19 data. Currently, 12.2 million full-genome SARS-CoV-2 sequences have been deposited in GISAID, and more than 100,000 new sequences are added every day using STGS platforms. The Illumina platform generated 75% of these sequences, Oxford Nanopore generated 23%, IonTorrent generated 1%, and Pacific Bioscience generated 1% [85]. The Illumina platform, which uses short reads (36–151 bp), has several advantages over other platforms, such as the ability to sequence the largest number of samples per run (up to 3072 in NovaSeq6000 with coverage of 300–600×), and its bioinformatic analysis flow is well established [86]. On the other hand, the Oxford Nanopore platform has the advantage of sequencing longer reads of up to 20 kb and can sequence viral RNA directly. IonTorrent and Pacific Bioscience are the least used platforms, perhaps due to their high startup cost and price per sequenced genome (Appendix A). The selection of a sequencing platform depends on the number of samples to be sequenced and on the monetary resources available, so SARS-CoV-2 sequencing must be performed with due consideration of the resources and capacities available in each laboratory [85,87]. In a few years, we will be able to use fourth-generation sequencing, also called in situ sequencing, which exploits second-generation NGS chemistry to read nucleic acid composition directly in fixed cells and tissues. This type of sequencing can be targeted when a specific primer is used to retrotranscribe a mRNA target or nontargeted when it uses random hexamers to reverse-transcribe mRNA. This approach opens a prospect for transcriptomic analysis of host-tissue interaction during SARS-CoV-2 viral infection [88].

### Direct RNA Sequencing

Most RNA-seq data have been obtained indirectly because current transcriptome analysis methods require RNA to be converted into cDNA prior to sequencing and the cDNA synthesis step introduces multiple biases and artifacts that interfere with its characterization in the quantification of the transcripts [89]. In SARS-CoV-2, sequencing requires an additional PCR step with 25–30 amplification cycles, which could introduce mutations derived from technical procedures [90]. In contrast, direct RNA-seq (DRS) is a powerful tool that allows for the characterization of the transcriptional landscape of viruses with complex genomes such as SARS-CoV-2 [91]. When reading the RNA genome directly, the complete sequence is obtained using long reads that do not require an assembly step or obtaining the precise identity of the nucleotide bases since it is not required to synthesize cDNA or perform PCR [92,93]. DRS is performed efficiently using the third-generation platforms Oxford Nanopore and PacBio [94]. Furthermore, the future of RNA-seq involves approaches such as parallel analysis of RNA structure (PARS) and the translatome. PARS helps to study the architecture and structure of viral RNA, allows for the identification of interactions with proteins, and focuses on SARS-CoV-2, which has helped to identify molecular mechanisms involved in spreading such as the interaction of the receptor binding domain with ACE2 receptors [95]. The translatome has shown great utility in determining RNA interassociation. With this approach, we can study beyond the expression level, and we can establish interactions between the virus and host, which can have clinical applications such as the design of small molecule inhibitors targeting viral replication pathways [96,97].

Since the world’s first COVID-19 case, STGS has been used to identify SARS-CoV-2. Currently, STGS continues to provide important information to public health officials and drug developers regarding vaccines because it has allowed: (1) monitoring of the transmission routes of the virus worldwide, (2) detection of mutations to prevent the spread of new types of variants, (3) identification of viral mutations that can cause false negatives in molecular diagnostic tests by RT–qPCR, and (4) detection of viral mutations that can impact the efficacy of vaccines and other treatments such as monoclonal antibodies [97]. The sequencing of SARS-CoV-2 genomes is performed in various settings, from less developed to more developed countries. Analysis of SARS-CoV-2 genomes has been shown to have and will continue to have an enormous potential to guide public health efforts around COVID-19 and should continue. This pandemic once again taught us that viral genomic surveillance is beneficial to public health.

## 5. The Landscape of Nucleic Acid-Based Therapies in SARS-CoV-2

Concerning nucleic acid-based therapy, RNA therapy has shown promising results in treating several human malignances [98,99,100], including viral diseases [101]. There are different approaches to targeting RNA in the human body: antisense oligonucleotides (ASOs) [102], microRNAs (miRNAs) [103], small interfering RNAs (siRNAs) [104], RNA vaccines [105,106], and CRISPR/Cas systems [101], among others. Nevertheless, due to poor RNA stability inside the body [107], inefficient delivery to the target site, and low bioavailability [108,109], RNA delivery remains one of the biggest challenges in RNA therapy [104,110]. Despite this obstacle, some therapies targeting RNA have been recently approved by the U.S. Food and Drug Administration (FDA); for example, spinal muscular atrophy (SMA) and hereditary transthyretin amyloidosis (hATTR) are two genetic diseases treated using an ASO (Nusinersen) [111,112,113] and an siRNA (Patisiran) [114,115,116], respectively. In addition to genetic diseases, RNA therapy can be applied to human infectious agents [117,118,119].

Due to the COVID-19 pandemic, interest in RNA therapy focused on viral diseases has grown. COVID-19 is caused by SARS-CoV-2, an RNA virus [120]. Its genome encodes several proteins, one of which is crucial to infection: the spike (S) glycoprotein, which is the major inducer of the host immune response [121] and has become an important target for vaccine development and therapy [122,123]. It binds to host receptors [124], promoting fusion between the viral envelope and host cell membrane [125]. RNA-based vaccines targeting the S glycoprotein have been a powerful tool for reducing the risk of infection [121]. Both FDA-approved RNA vaccines by Pfizer/BioNTech and Moderna contain mRNA to express the S glycoprotein [126] for immunization. The Pfizer/BioNTech [127] and Moderna [128] vaccines have been confirmed to induce the production of anti-S IgG antibodies with neutralizing activity against SARS-CoV-2 and to reduce the risk of infection and the development of acute symptoms. Sequelae and side effects in patients after SARS-CoV-2 infection are broad [129] and include fatigue [130], metabolic disorders [131,132], kidney injury [133], cardiovascular disease [134], and neuropsychiatric disorders [135,136], among others [137]; thus, to reduce further development of sequelae after infection, it is important to identify efficient and specific therapies.

### 5.1. Small Interfering RNAs

These noncoding double-stranded transcripts are ~20 nucleotides long and mediate the specific suppression of transcripts [138,139] by binding to the RNA-induced silencing complex (RISC) [140]. The single-stranded siRNA guides RISC to its target RNA, leading to gene silencing [141]. In the pandemic caused by SARS-CoV in 2003, siRNAs effectively reduced viral replication in infected cells [142,143,144] and in in vivo models [145]. Regarding SARS-CoV-2, in a study in which primary human tracheal cells (hpTCs) expressing the S glycoprotein were treated with siRNAs [146], a significant reduction in this protein was reported, cell viability and proliferation remained unchanged, and modified siRNAs with cholesterol moieties were used for cells to directly uptake siRNA from the medium, a useful approach for future therapeutic applications in which some delivery methodologies may cause toxic effects [107,147]. In the search for lower-toxicity delivery systems, a lipid nanoparticle was used to deliver siRNAs targeting RNA-dependent RNA polymerase (RdRp), helicase (Hel), and the 5′ untranslated region (5′-UTR) in the SARS-CoV-2 genome using a K18-hACE2 mouse model [148], and a 90% reduction in viral replication was observed. COVID-19-associated symptoms were lower in treated mice, which also exhibited higher overall survival, demonstrating that siRNAs are a useful therapy against SARS-CoV-2. In this study, the siRNA delivery was made through a formulation described previously [149] in which the polyethylene glycol (PEG)2000–C16 ceramide conjugate was prepared using the hydration of freeze-dried matrix (HFDM) method to form liposomes; furthermore, intravenous delivery of this conjugate proved successful delivery of siRNA in the lung epithelium and achieved gene silencing in these cells [150]. Nevertheless, the precise transfection mechanism into the cells remains to be described. Using another in vivo model, an siRNA targeting the RdRp gene that significantly reduced viral replication and lung inflammation in Syrian hamsters was reported [151]. This siRNA administration was performed by inhalation, a tool that could be a promising way to deliver drugs for respiratory diseases; in this case, a cationic dendrimeric peptide (KK-46) proved to be successful in the intracellular delivery of siRNA with very low toxicity. Therapeutic siRNAs have a confirmed efficacy against SARS-CoV-2; nevertheless, clinical studies are needed to support siRNA therapy as safe and efficient for COVID-19 in humans.

### 5.2. MicroRNAs (miRNAs)

In contrast to siRNAs, miRNAs are single-stranded RNAs with a length of 19–25 nucleotides [152] that can regulate not only one target, but may have different targets [153,154]; nevertheless, both noncoding RNAs share a similar mechanism of action through their association with RISC and Dicer for post-transcriptional inhibition of mRNA [155,156]. Such miRNAs have shown to play key roles during human diseases such as cancer [157,158] and in infectious diseases [159,160]. In SARS-CoV-2 infection, expression of more than 70 miRNAs were deregulated in peripheral blood [161], suggesting the importance of these noncoding RNAs in the infectious process. They have also been proposed as potential biomarkers of the severity of COVID-19 [162,163], as well as for therapeutic approaches to SARS-CoV-2 infection [164,165].

### 5.3. Antisense Oligonucleotides

ASO-based therapy has been successfully used for human viral diseases [166,167]; in general, ASOs are single-stranded DNA sequences of ~20 nucleotides in length that bind to complementary RNA via Watson–Crick base pairing [168,169,170]. ASOs can modify RNA functions through mechanisms that depend on their chemical structure and modifications [170,171]. ASOs can interfere with mRNA translation [172,173] and splicing [174] but can also recruit RNase H1 and H2 to induce RNA degradation [81,175]. Several ASO-based therapies depend on these mechanisms to maintain low levels of the target sequence [176,177,178]. Therapies against human viruses such as hepatitis B [179,180], Ebola [181], influenza A [182], and West Nile virus [183], among others, are being developed and have demonstrated high specificity and efficacy, suggesting a promising landscape for ASO-based therapies.

Regarding SARS-CoV-2, in a study in which ASOs were designed using in silico tools to disrupt the interactions between SARS-CoV-2 5′-RNAs and host proteins needed to regulate the viral infection cycle [184], in vitro experiments on the human cell line Huh7.5.1 reported an ~50% decrease in the SARS-CoV-2 RNA yield in treated cells. They also designed ASOs that targeted structurally conserved regions of SARS-CoV-2 RNA (ORF1 and N) that reduced the viral infection ratio by 50% in the human cell line Caco-2, which was derived from human colon carcinoma. Cells were transfected with ASOs using lipofectamine; the transfection mechanism was the formation of cationic liposomes that enabled the internalization of anionic ASOs. Then, in contact with the cell membrane, liposomes created endosomes to enter the cell [185]. The innovative in silico tool they proposed for ASO design succeeded in targeting different regions of the viral genome and reduced the viral yield and infection.

Peptide nucleic acids (PNAs) are another type of ASO that consist of a polypeptide backbone with nucleic acid bases attached as side chains and are used as DNA/RNA analogs to inhibit transcription and translation [186]. In a study conducted by Ahn and collaborators [187] using programmed −1 ribosomal frameshifting (−1 PRF) in the SARS-CoV genome, a region controlling the synthesis of viral replicase proteins [188] was targeted using PNAs in HEK293 cells; inhibition of viral replication was observed compared to the control, suggesting that -1 PRF is an important target for SARS-CoV treatment. Cells were transfected using FuGENE, a cationic polymer that creates complexes with anionic ASOs. The complexes could create endosomes in the cell membrane to penetrate the cell [185,189]. Even though ASOs have been demonstrated to be efficient and specific in targeting SARS-CoV-2 in vitro and in vivo, further experiments are needed to confirm their potential use as therapeutic agents in humans against SARS-CoV-2.

### 5.4. CRISPR-Cas

Clustered regularly interspaced short palindromic repeats (CRISPR) and CRISPR-associated proteins (Cas) systems are an adaptive immune system in prokaryotes [190,191,192] that was first described in the late 1980s and early 1990s [193]. Currently, the therapeutic gene-editing applications of this technology are a promising tool to treat several human malignances [194,195,196] and viral diseases [197,198]. Among CRISPR-Cas effectors, two recently discovered effectors are specific to targeting single-stranded RNA (ssRNA), type VI-A (Cas13a) and type VI-B (Cas13b) [199], which were the first discovered among CRISPR-Cas systems that specifically targeted RNA [175,200,201]. One of the RNA viruses previously targeted by CRISPR-Cas13 is HIV-1. Yin and collaborators [119] observed a strong inhibition of HIV infection in human HEK293T cells, reduced viral RNA and protein levels, and reduced viral DNA integration after infection with mature HIV-1 viral particles compared to control treatment.

Regarding SARS-CoV-2, Abbott and collaborators [202] reported a CRISPR-Cas13-based strategy called prophylactic antiviral CRISPR in human cells (PAC-MAN), which revealed a group of 6 crRNAs able to target 91% of all sequenced coronaviruses and a group of 22 crRNAs that could target 100% of the sequenced coronaviruses. In experiments on the human lung epithelial A549 cell line using reporters of synthetic fragments of SARS-CoV-2 RdRP or nucleocapsid (N) genes fused to GFP, pools of four of the predicted crRNAs were used to target these constructs, resulting in a 70% reduction in the reporter gene expression. The use of a pool of crRNAs could be helpful in the case of a live infection in which mutation or the presence of viral variants could escape from a single crRNA. In contrast to this virus-free model, in a study conducted by Fareh and collaborators [203], CRISPR-Cas13b was used to target the mRNA encoding the N protein in replication-competent particles of SARS-CoV-2 using the Calu-3 cell line derived from human lung adenocarcinoma. After viral infection, the silencing efficiency, as measured by a tissue culture infectious dose that inhibited 50% of the virus growth in an infectivity assay (TCID50), showed that replication was significantly lower compared to the control, even at the longest time measured (48 h). Similar results were observed in Vero cells, a nonhuman mammalian cell line in which pools of four different crRNAs were used to target several regions of the SARS-CoV-2 genome; 80–90% viral suppression was achieved with all tested crRNAs, as well as suppression of the SARS-CoV-2 variant D614G and B.1.1.7, known as the UK variant. In this study, the cell line expressing Cas13d was transduced by lentiviral infection with the pooled crRNAs. The lentiviral transduction mechanism was based on the viral infection of the cells by the virion; the virion could enter the cell by endocytosis or by binding to specific receptors, then the nucleic acid contained inside the virion was released inside the cell [204,205].

In another study using a replication-competent SARS-CoV-2 virus [206], CRISPR-Cas13a crRNAs were designed to target RdRp and N genes in the Vero E6 cell line. A combination of two crRNAs targeting the N gene reduced 72% of cell death in infected cells, while the mRNA of RdRp and N genes exhibited a reduction of 93.7% and 94.1% in copy number, respectively. After successful in vitro assays, in vivo experiments were performed on Syrian hamsters using a crRNA targeting the N gene. Administration of mRNA encoding Cas13a and crRNA formulated with poly-beta amino ester, which was suitable for delivery through inhalation, was performed using a cone nebulizer; 20 h later, hamsters were infected with SARS-CoV-2. Six days later, a 57% decrease in the viral N mRNA copy number was observed compared to the control, suggesting CRISPR-Cas13a as a potential antiviral therapy against SARS-CoV-2. The in vivo delivery of mRNA-encoding CRISPR-Cas13a and crRNAs was performed using hyperbranched poly (beta amino esters) (hPBAEs); these compounds create polyplexes with the anionic RNAs that allow endosome formation in the cell membrane to penetrate the cell [207].

CRISPR-Cas13 systems have been demonstrated to be efficient for SARS-CoV-2 RNA knockdown not only in in vitro virus-free models, but also when using replication-competent viral particles in human cell lines and in vivo models (Figure 3). The approach to using CRISPR-Cas13 goes beyond therapy against COVID-19; due to its specificity and flexible design against different genomic targets, it could be used to treat different human viral diseases and represents a powerful weapon against future pandemics.

## 6. Precision Medicine and Precision Public Health in SARS-CoV-2

As a concept, PM refers to a deep understanding of the disease with the aim of establishing improved targeted diagnostic and therapeutic interventions [208]. PM is based on the integration of patient information including lifestyle, environmental exposure, serum biomarkers, imaging studies, and gene analysis, which allows for the best selection of treatment [209]. PM is increasingly recognized as synonymous with a patient-centered approach that is driven by technology [209,210]. The development and sophistication of molecular biology techniques enable the establishment and utility of PM [209,211]. With the emergence of genomics methodologies, other similar approaches have emerged such as transcriptomics, epigenetics, proteomics, metabolomics, and other omics [211,212]. These approaches provide an enlarged resolution of each clinical context from different points of view. On the other hand, PPH is essentially similar to PM; however, PPH interventions are directed at the population rather than individuals [212,213,214]. Many definitions have been proposed to describe PPH since the term was first coined in 2011. PPH can be understood as “a novel approach that uses big data science technology to improve population health and reduce health disparities”. PPH seeks to provide the right intervention to the right population at the right time [212]. Recently, the concept of PPH and its scope have re-emerged due to the SARS-CoV-2 pandemic [213]. In this context, its applications range from modernizing public health surveillance to the early identification and tracking of new outbreaks and rapid genomic characterization and population susceptibilities [214].

PPH’s objective is to identify the earliest indicators of disease and to characterize the pathogen or etiological factor and transmission kinetics [215] to facilitate identification of more effective targeted drugs and lifestyle interventions [212,213,216]. PPH must be able to be applied globally but faces the challenge of health disparities in terms of access and quality of health care, influencing politics and economics, which are directly related to health care [213,217,218]. In particular, the COVID-19 pandemic provides a fertile field for the evolution of PPH, an opportunity to highlight its importance and bring together new tools and technologies to complement traditional medical and public health approaches. PPH has already provided valuable fruits of its application in the COVID-19 pandemic; i.e., Sharon Greene and colleagues observed how the COVID-19 pandemic spread across New York City; they were able to map outbreaks as they unfolded across individual neighborhoods almost in real time [216]. This gave the team the opportunity to redistribute testing and protective supplies such as masks and gloves to the right places [216]. Additionally, it is remarkable that the identification of the population susceptible to severe forms of disease can be achieved not only via the clinical profile, but also via the molecular characterization and genotyping of the different inflammatory responses. The identification of these susceptibilities has provided a guideline to establish different treatment strategies [219]. Another example is the identification of SARS-CoV-2 variants with clinical relevance; i.e., the Omicron outbreak [215]. Rapid characterizations of new outbreaks at the geographic and genomic levels [220] have made it possible to better allocate resources [221,222] as well as to prepare health care providers and facilities for the expected increase in the number of cases [222]. Therefore, the development of vaccines is a breakthrough in the fight against SARS-CoV-2 [223,224]. Various candidate vaccines including inactivated viral vaccines, live attenuated vaccines, nucleic acid vaccines, viral-vectored vaccines, and protein or peptide subunit vaccines are being rapidly developed, tested, and granted approval for emergency use [225]. Currently, nasal-spray vaccines are now the new hope of many research groups that are working on new kinds of inoculation. Rather than relying on injections, these use sprays or drops administered through the nose or mouth that aim to improve protection against the SARS-CoV-2 virus [226]. There are important theoretical advantages attributed to mucosal vaccines compared to conventional injected vaccines. The mucosal site of inoculation favors the activation of immune cells in the nasal mucosa and respiratory tract, thus allowing the establishment of a more rapid and effective immune response, which could provide better protection against asymptomatic or mild infections [226]. This represents an important strategy to mitigate the SARS-CoV-2 pandemic.

The success of these interventions depends on integrative collaboration among the private health sector, public health institutions, primary care providers, governments, and communities [213]. PPH is now shaping the way humans interact with each other, and its precepts are guiding our lives through the pandemic [212,217,227]. Figure 4 shows how the PPH approach has served as a platform for designing and targeting these interventions summarized in eight key points, from the detection of the first cases of an unknown respiratory disease, to the integration of the clinical picture and identification of risk factors [228] to molecular characterization and the development of accessible diagnostic tests to develop effective drugs [229]. Similarly, artificial-intelligence-based models have enabled the targeted search and identification of therapeutic targets and the ability to redirect drugs approved for humans. Such is the case of baricitinib, which has shown a significant benefit in patients with high oxygen requirements [230].

This approach has accelerated the transition from the practice of conventional medicine to precision public health-based medicine. A little more than two years after the pandemic, it seems that telemedicine is here to stay, not as a luxury but as a necessity to ensure access to health care for everyone. Likewise, the safety and development of messenger RNA-based vaccines have laid the groundwork to prevent not only COVID-19, but also other diseases that impact humanity [238]. The development of SARS-CoV2 vaccines has indirectly favored the development of immunotherapy by characterizing T-cell, B-cell, and myeloid-cell responses. The implication of this knowledge has vast implications, especially in cancer treatment. Currently, this type of strategy can be applied in future SARS-CoV-2 outbreaks to prevent pandemics [225]. Additionally, artificial intelligence (AI) is changing drug development, and the best targets will now be unveiled [239]. Despite current and emerging challenges, there is no better time to turn public health into PPH than now.

## 7. Conclusions

Transcriptomics analysis during the SARS-CoV-2 pandemic has allowed for the study of the molecular mechanisms that are associated with the infection process and pathogenesis of acute respiratory disease due to COVID-19 because it allows us to identify genes whose expression is deregulated in response to the infection process using highly specific studies such as those provided by new spatial transcriptomics technologies. Furthermore, analysis of the transcripts produced by SARS-CoV-2 allows us to understand the composition of its genome and its ability to adapt by generating variants that allow it to increase its transmissibility between hosts, which can also increase its pathogenicity and the generation of variants associated with the development of severe respiratory diseases. Likewise, rapid advances in single-cell sequencing technologies have opened a new era for biomedical research to identify new biomarkers and drug targets, greatly promoting the development of more effective means for disease diagnosis, molecular subtyping, and medical treatment. On the other hand, pandemics are large-scale outbreaks of infectious diseases in which global morbidity and mortality can increase considerably, resulting in significant effects on individual countries, the global economy, and politics. The epidemiological record recently suggested that pandemics have increased over the past century due to increased continental and intercontinental travel as well as movement of persons. Consequently, the likelihood of future outbreaks is increasing, driving the need to find better and faster strategies to treat viral diseases, such as nucleic acid-based therapy, which, due to its high target specificity, is a promising target therapy to overcome future infectious diseases such as SARS-CoV-2.

Notably, current medicine has a series of commitments in terms of humanity that have been accentuated by the global crisis caused by SARS-CoV-2. Human decisions play a crucial role in the diagnosis and treatment of an emerging clinical entity, but they have limitations with respect to optimizing and distributing resources and vaccination. This is an opportune moment for the term PPH to be revitalized given a massive analysis (transcriptomics, epidemiological, clinical, biochemical, genomics, genetics, and proteomics, among others) to generate predictive algorithms within the reach of anyone using digital platforms that facilitate individualizing life-and-death decisions. Lastly, there is a need to create synergies among omics sciences and RNA-based therapies to enable the development of advanced PPH-focused systems that facilitate the elicitation of relevant clinical strategies for rapid decision making and support the personalized management of SARS-CoV-2-infected patients in case a new wave of this virus should emerge.

## Figures and Tables

**Figure 1 ijms-23-11058-f001:**
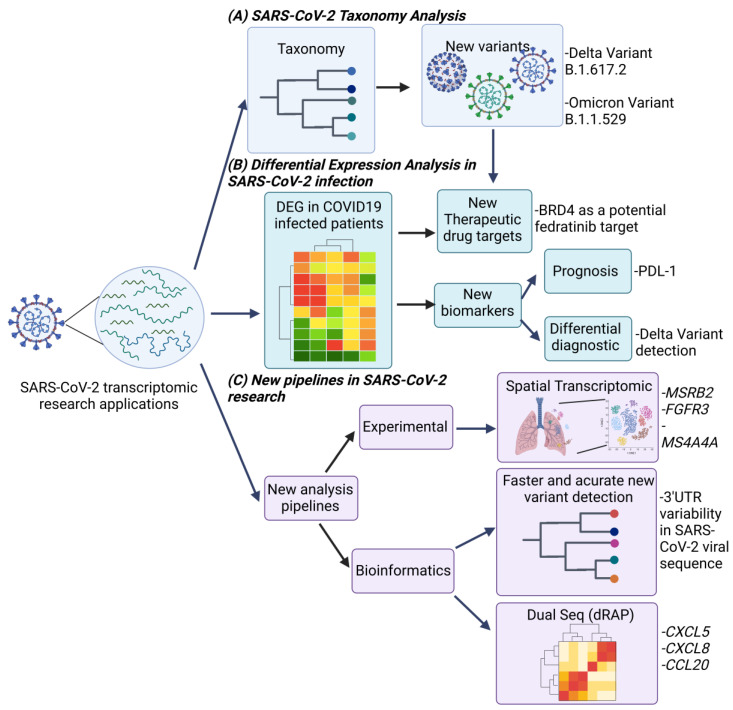
Research applications of SARS-CoV-2 viral infection using transcriptome analysis. Analysis of the transcriptome of viral infection by SARS-CoV-2 has focused primarily on three areas: (**A**) the investigation of the biological nature and taxonomy of the virus. The study of taxonomy has focused primarily on the characterization of the SARS-CoV-2 genome and the mutations it presents, which has led to the identification of new variants of the virus. (**B**) Analysis of differentially expressed genes in patients with COVID-19 infection. Analysis of the transcriptome of patients who are hosts of this virus has allowed the identification of differentially expressed genes that deregulate the intracellular signaling pathways involved in infection and viral pathogenesis, as well as their relationship with the presence of variants of the virus, which, together with the characterization of variants, has allowed the identification of new therapeutic targets, as well as new molecular diagnostic biomarkers, which may be useful in the prognosis of patients with severe disease and differential diagnosis of the variants of the virus, as well as the distinction of respiratory pathologies of different etiologies. (**C**) The development of new methodologies for transcriptome analysis. To optimize research in these areas, new experimental strategies have been developed that allow for obtaining more information about viral pathogenesis such as spatial transcriptomics, as well as new bioinformatics strategies such as the development of new workflows that optimize the detection of variants during sequencing, as well as the joint analysis of the host transcriptome and the virus by Dual-Seq, which optimizes resources for the study of SARS-CoV-2.

**Figure 2 ijms-23-11058-f002:**
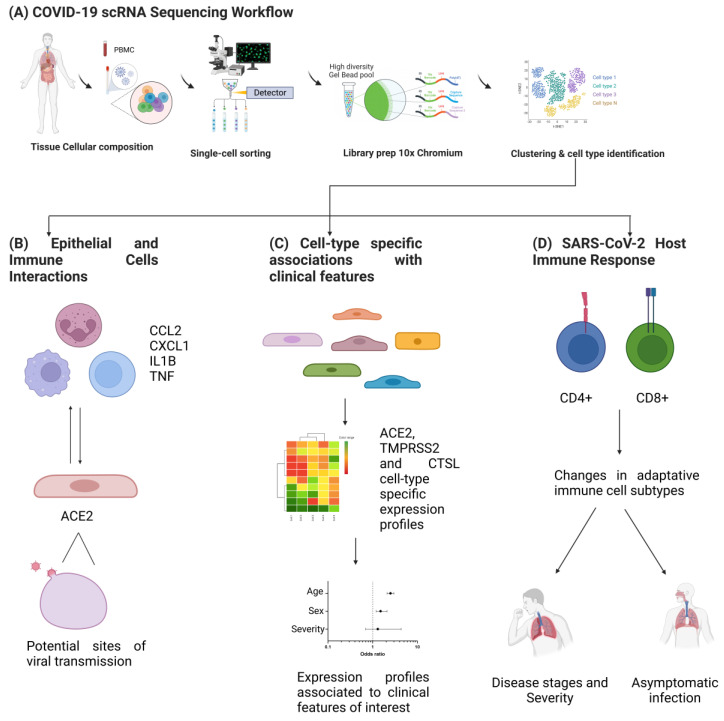
Single-cell RNA-seq applications for COVID-19 clinical research. (**A**) A typical scRNAseq workflow consists of sampling followed by single-cell sorting and library preparation, which involve the ligation of adapters and sample indices, allowing for the pooling and sequencing of multiple libraries on a next-generation short read sequencer and the subsequent cell type identification by transcriptomic analysis. This approach has multiple applications in COVID-19 clinical research, such as: (**B**) the characterization of new cell-type-specific interactions, particularly among epithelial and immune cells, mediated by ACE2 receptor and pro-inflammatory elements such as CCL2, CXCL1, IL1B, and TNF, which together are involved in viral transmission and could allow the identification of novel potential sites for viral transmission; (**C**) the identification of cell-type-specific expression profiles associated with clinical features of interest, such as the characterization of cell subpopulations by their expression of ACE2, TMPRSS2, and CTSL, which together have been associated with clinical variables such as age, sex, and severity of disease in COVID-19 patients, allowing for the identification of prognostic clinical biomarkers; and (**D**) finally, scRNA transcriptomics in COVID-19 allows for the identification of biomarkers for differential diagnostic and prognostic tests using the characterization of transcriptomic changes in CD4+ and CD8+ cell subtypes, which mediate the host adaptative immune response, regulate viral infection and pathogenic processes, and are associated with disease stages and COVID-19 severity.

**Figure 3 ijms-23-11058-f003:**
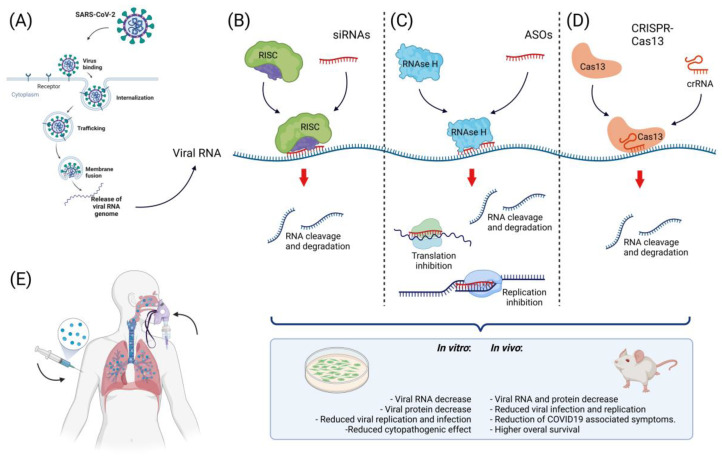
Approaches to nucleic acid-based therapy against SARS-CoV-2. (**A**) Scheme of viral infection with SARS-CoV-2. After viral binding to cell receptors, the viral particle is internalized and then releases viral RNA to the cytoplasm, where viral components are synthesized. At this point, free viral RNA in the cytoplasm can be targeted. (**B**) One of the strategies for targeting viral RNA inside the cell is the use of siRNAs; these small molecules (~20 nt) regulate RNA degradation through their interaction with the RISC complex. The activated RISC complex binds to its target, resulting in RNA cleavage and degradation; the use of siRNAs targeting SARS-CoV-2 has been successful in vitro and in vivo. (**C**) Another approach to targeting this virus is the use of ASOs; these molecules mediate RNA cleavage by recruiting RNAse H and interfere with replication, transcription, and transduction, resulting in decreased viral RNA and protein levels. ASOs have been successfully used to target SARS-CoV-2 in vitro. (**D**) The use of genome-editing tools has increased the possibilities for SARS-CoV-2 treatment, while the use of CRISPR-Cas13 has been successful in targeting this virus in vitro and in vivo, resulting in a reduction in viral RNA and protein, a decrease in viral infection, and a decrease in COVID-19-associated symptoms. (**E**) For the successful targeting of viral RNA, the use of an efficient delivery method that guarantee the proper release of the therapeutic nucleic acid to the cell is important. For the internalization of negatively charged nucleic acids into the cell, the use of cationic compounds and lipid-based delivery methods have been successfully used for this particular therapy via intravenous or inhalation delivery.

**Figure 4 ijms-23-11058-f004:**
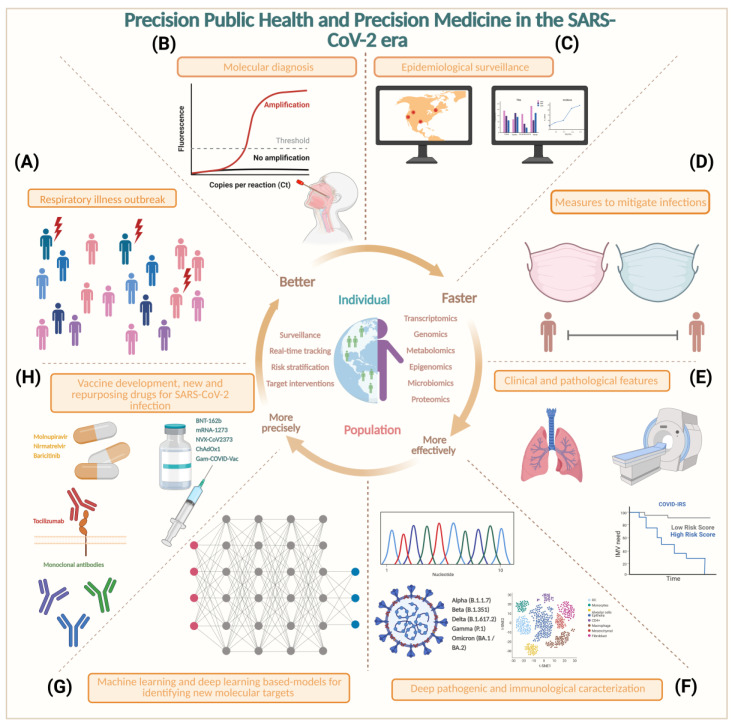
Precision public health and precision medicine in the SARS-CoV-2 era. Conceptual framework showing the integrated relevance of PPH and PM in response to the emergence of the COVID-19 pandemic. (**A**) Respiratory illness outbreak. Cases of a new respiratory illness began to increase. (**B**) Molecular diagnosis. Access to accurate tests allowed the confirmation of cases, and RT–qPCR is the gold standard (sensitivity 93% and specificity 100%) [231,232]. (**C**) Epidemiological surveillance. Clinical and population data are integrated by the epidemiological surveillance centers of the region. (**D**) Measures to mitigate infections. Social distancing and mask use are implemented to reduce contagions. (**E**) Clinical and pathological features. Eventually, with the increase in the number of cases, it is possible to build a clinical and epidemiological profile, identifying risk factors and prognostic tools [233]. (**F**) Deep pathogenic and immunological characterization. In an approach driven by technological development, it was possible to perform a molecular characterization of the viral agent and to establish its infectious nature, as well as the susceptibilities of populations in cellular, transcriptomic, genomic, and epigenomic approaches [228]. (**G**) Machine learning (ML)- and deep learning (DL)-based models for identifying new molecular targets. The complementary method to conventional health care has been implemented based on AI such as ML and DL to identify patterns of susceptibility and potential targets for drug development and vaccine development [234]. (**H**) Vaccine development and new and repurposed drugs for SARS-CoV-2 infection. Ultimately, with the advent of vaccines [235] and new drugs [236,237] useful against COVID-19, they open the door to evaluating their efficacy on an ongoing basis with the goal of targeting the right intervention to the right population at the right time.

**Table 1 ijms-23-11058-t001:** Overview of software used for scRNAseq data analysis.

Software	Category	Reference *
TopHat2 v2.1.1	Read mapping	https://ccb.jhu.edu/software/tophat/index.shtml Cole Trapnell, Lior Pachter and Steven Salzberg at the University of Maryland, UC Berkeley, USA [58]
STAR v2.7.10a	Read mapping	https://github.com/alexdobin/STAR Alexander Dobin, Cold Spring Harbor Laboratory, NY, USA [59]
HISAT2 v2.2.2	Read mapping	http://daehwankimlab.github.io/hisat2/ Kim, D., Langmead, B. & Salzberg, S. Baltimore, Maryland, USA [60]
Cufflinks v0.17.3	Expression quantification	http://cole-trapnell-lab.github.io/cufflinks/ Cole Trapnell, et al. University of Maryland, College Park, Maryland, USA [61]
RSEM v1.1.17	Expression quantification	https://github.com/deweylab/RSEM Bo Li and Colin N Dewey. University of Wisconsin-Madison, Madison, WI, USA [62]
StringTie v2.1.0	Expression quantification	https://ccb.jhu.edu/software/stringtie/ MIT License Johns Hopkins University, Baltimore, Maryland, USA [63]
Cell Ranger v3.1.0	Align reads, generate feature-barcode matrices, perform clustering and other secondary analyses	https://github.com/10XGenomics/cellranger © 2022 10x Genomics. California USA [52]
Seurat v4.0	Filter based on RNA, number of detect genes, and number of total UMIs,	https://satijalab.org/seurat/ Hao, et al., Center for Genomics and Systems Biology, New York University, New York, USA [64]
Scrublet v0.2.1	Single-cell remover of doublets	https://github.com/swolock/scrublet Samuel Wolock, MIT [55]
SoupX v1.2.2	estimation and removal of cell free mRNA contamination	https://github.com/constantAmateur/SoupX Matthew Young. Wellcome Trust Sanger Institute, Cellular Genetics, Wellcome Genome Campus, Hinxton, CB10 1SA, UK [57]
Waterfall	Dimensionality reduction	https://doi.org/10.1016/j.stem.2015.07.013 Jaehoon Shin et al. Johns Hopkins University School of Medicine, Baltimore, MD 21205, USA [65]
TSCAN v1.0	Dimensionality reduction	https://github.com/zji90/TSCAN Zhicheng Ji, Hongkai Ji. Johns Hopkins Bloomberg School of Public Health, Baltimore, MD, USA [66]

* Recent program versions were considered. 13 September 2022 was taken as the latest date to access.

**Table 2 ijms-23-11058-t002:** Clinical studies related to COVID-19 and scRNAseq.

Study	Status	Interventions	Outcome Measures	Enrolled Patients	Locations	Identifier
COVID-2019 Vaccine Immune Response Based on Single Cell Multi-Omics	Recruiting	Biological: recent vaccination	Changes in classification of human peripheral blood mononuclear cells	50	China	NCT04871932
Virological and Immunological Monitoring in Patients (Suspected of/Confirmed With) COVID-19	Active, not recruiting	Procedure: blood draw Procedure: bronchoalveolar lavage Procedure: SARS-CoV-2 swabs	Identification of cytokines and chemokines associated with COVID-19 severity and outcome Identification of cellular subsets that can predict COVID-19 severity and outcome SARS-CoV-2 sequencing	109	Belgium	NCT04904692
COVID-19 in Baselland: Investigation and Validation of Serological Diagnostic Assays and Epidemiological Study of SARS-CoV-2 Specific Antibody Responses COVID-19	Recruiting	Diagnostic test: blood draw Diagnostic test: fingertip tests for POC assays Diagnostic test: saliva collection Diagnostic test: collection of swabs	Qualitative method validation (yes/no) Quantitative method validation (antibody concentrations) Immune cell repertoire sequencing	550	Switzerland	NCT04483908
Myeloid Cells in Patients with COVID-19 Pneumonia	Not yet recruiting	Other: blood sampling Other: nasal brushing	Myeloid cell subpopulation phenotype Myeloid cell functions Myeloid cell transcriptomic and proteomic study.	120	France	NCT04590261

## Data Availability

Not applicable.

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
