# Peer review of "Transcriptomics and RNA-Based Therapeutics as Potential Approaches to Manage SARS-CoV-2 Infection"

_ijms, 2022, doi:10.3390/ijms231911058_

Round 1

Reviewer 1 Report

The authors have compiled a very relevant review of the growing area of transcriptomics and RNA-based therapies. This has good impact in the current requirement for rapid diagnostics, personalised and novel therapies, and the threats of long-covid. Some further edits are required before acceptance.

The abstract can be bolstered as it comes across as a bit too minimal in information and message.

Several sections do not flow well grammatically and there is an overuse of linking/transition words (eg. “finally,” and “in summary,”). A thorough edit of the writing and check for grammatical errors is required.

A description of In situ/4th gen sequencing can be expanded on at line 346.

Line 397 – is this statement repeated? Perhaps the topic can appear earlier in the manuscript and then expanded on in this section.

Delivery mechanisms for each of the RNA therapies can be expanded in each of the subsections.

A small section on miRNA can be included after the siRNA section.

Section 6 needs a thorough revisiting. Enable that the connections that are trying to be made between precision medicine and the topic of the article are clearly made. Greater time perhaps should be spent exploring the importance of the discussed techniques and the development of mRNA-based vaccines.

Line 625 – this statement is slightly ambiguous. Restructure and rephrase with a clearer message.

I hope that these comments will help in your manuscript reaching publication. Kudos on compiling this work.  

Author Response

Response to Reviewer 1 Comments

Point 1: The abstract can be bolstered as it comes across as a bit too minimal in information and message.

Response 1: We agree, the abstract has little information or message. Therefore, some specifications were added in the abstract to mention transcriptomics techniques used for molecular diagnosis of SARS-CoV-2. Additionally, we mentioned the molecular techniques used in RNA-Based therapeutics, such as CRISPR-CAS, ASOs, and siRNAs, please see lines 27-28 and 30-32. Here, we cannot add more information because the abstract is limited to 200 words according to the journal specifications, so we try to be as specific as possible.

Point 2: Several sections do not flow well grammatically and there is an overuse of linking/transition words (eg. “finally,” and “in summary,”). A thorough edit of the writing and check for grammatical errors is required.

Response 2: We agree, several sections do not flow well grammatically and there is an overuse of linking/transition words (eg. “finally,” and “in summary,”).  A deep revision of the text was carried out and corrections were made with the objective that the manuscript was continuous in wording. The connectors "finally," and "in summary" were avoided in the manuscript. Please see lines 48–50, 70, 120–124, 127–133, 142, 146, 157–162, 167–168, 176, 184, 221, 281, 395, 556, 603, 619, 641, 689, 702, and 709.

Point 3: A description of In situ/4th gen sequencing can be expanded on at line 346.

Response 3: We agree, an expanded description of fourth generation sequencing is included, please see lines 353-359.

Point 4: Line 397 – is this statement repeated? Perhaps the topic can appear earlier in the manuscript and then expanded on in this section.

Response 4: In this case, we tried to give a brief description of RNA therapeutics in the introduction (lines 57-60) and later, in the "The landscape of nucleic acid-based therapies in SARS-CoV-2" the same concept was taken up again, but the manuscript gives a detailed description of RNA Therapeutics and the genes that have been targeted for the development of new vaccines based on viral mRNA. Therefore, it is emphasized that the SARS-CoV-2 genome corresponds to an RNA virus and one solution is to use different strategies based on RNA molecules that include the use of ASOs, siRNAs and CRISPR-Cas systems. Although it may seem repetitive, this topic is mentioned only in general terms in the introduction and is described in detail later on.

Point 5: Delivery mechanisms for each of the RNA therapies can be expanded in each of the subsections.

Response 5: We agree, delivery mechanisms for each RNA therapy in each subsection are included in the manuscript. Please see lines 441-446, 450-451, 487-490, 499-502, 536-541, 552-555.

Point 6: A small section on miRNA can be included after the siRNA section.

Response 6: We agree, a small section on miRNA was included after the siRNA section. Please see lines 456-467.

Point 7: Section 6 needs a thorough revisiting. Enable that the connections that are trying to be made between precision medicine and the topic of the article are clearly made. Greater time perhaps should be spent exploring the importance of the discussed techniques and the development of mRNA-based vaccines.

Response 7: We agree, section 6 was thoroughly revised. An attempt was made to connect the main topics including Public Precision Health, transcriptomics and RNA-based therapeutics. Some specific examples were given, and an attempt was done to make the connection suggested by Reviewer 1. Please see lines 611-615, 620-632, 641-644, 671-675.

Point 8: Line 625 – this statement is slightly ambiguous. Restructure and rephrase with a clearer message.

Response 8: We agree, this statement is slightly ambiguous. The paragraph was restructured, and specific emphasis was placed on the pandemic concept, as well as the consequences that this type of events have globally. Please see lines 692-700.

In the hope that we have responded to all the suggestions made by the Reviewer#1, we sincerely hope that the new modifications made to the manuscript are in accordance with what was requested. Thank you for the time invested in reviewing our work.

Reviewer 2 Report

The authors propose an exhaustive general review on using Transcriptomics and RNA-Based therapeutics in the management of coronaviruses approach in general and COVID-19 infection in particular. The extensive literature citations and scientific opinion-based approach makes a good scholastic paper with interesting conclusions of the type "what we have learnt and what we should do".

However, I would make some more references regarding immediate and proximal clinical approaches in this field (very good example demonstrated in figure 4). 

Overall, a well-written scholastic paper that can contribute to the general knowledge in the field.

Author Response

Response to Reviewer 2 Comments

Point 1: However, I would make some more references regarding immediate and proximal clinical approaches in this field (very good example demonstrated in figure 4).

Response 1: We agree, as suggested by Reviewer#2 sentences were included to explain immediate and proximal clinical approaches in this field. Please see lines 611-615, 620-632, 641-644, 671-675.

In the hope that we have responded to all the suggestions made by Reviewer#2, we sincerely hope that the new modifications made to the manuscript are in accordance with what was requested. Thank you for the time invested in reviewing our work.
